# CarbaDetector: a machine learning model for detecting carbapenemase-producing Enterobacterales from disk diffusion tests

Linea Katharina Muhsal [1,2], Cansu Cimen[1,3], Janko Sattler [4,5], Lisa Theis [1], Oliver Nolte [6], Laurent Dortet [7,8], Rémy A. Bonnin[7,8], Adrian Egli[6] & Axel Hamprecht [1,9] ✉

Carbapenemase-producing *Enterobacterales* (CPE) are considered among the highest threats to global health by WHO. Their detection is difficult and time-consuming. We developed a random-forest machine learning (ML) model, CarbaDetector, to predict carbapenemase production from inhibition zone diameters of eight antibiotics, using 385 isolates for training with whole genome sequencing as reference. Validation on two external datasets (A = 282, B = 518 isolates) shows high performance: sensitivity/specificity are 96.6%/84.4% (training), 96.3%/86.1% (A), and 91.2%/87.0% (B, five antibiotics). In contrast, the algorithms of EUCAST and the Antibiogram Committee of the French Society of Microbiology (CA-SFM) exhibit lower specificity (8.2% and 40.1%, respectively on the training dataset). In this work, we show that CarbaDetector, available as a web-app, reduces unnecessary confirmatory testing and accelerates the time to result. This approach offers high sensitivity and improved specificity compared to standard algorithms and has the potential to improve CPE detection, especially in resource-limited settings.

Carbapenemase-producing *Enterobacterales* (CPE) are a global health threat and are listed in the "critical group" of the WHO's 2024 Bacterial Priority Pathogens List[1]. CPE accounted for ~6.5% (34,934) of 541,000 deaths associated with bacterial antimicrobial resistance in the WHO European region in 2019, making them one of the most significant multidrug-resistant Gram-negative bacteria[2]. Infections caused by CPE are a challenge due to few therapeutic alternatives available[3]. Timely and accurate detection and confirmation of CPE is crucial not only to implement necessary infection control measures to help preventing the spread of

carbapenemase genes and CPE outbreaks, but also to initiate adequate treatment.

The European Committee on Antimicrobial Susceptibility Testing (EUCAST) recommends the combination of a two-step procedure to detect CPE, with an initial screening step, followed by at least one confirmatory test[4]. The screening step involves determining the minimum inhibitory concentration (MIC) or the disk diffusion zone diameter using meropenem or ertapenem[4]. Confirmatory tests involve combination disk tests[5], molecular diagnostics[6], colorimetric tests such as CarbaNP[7,8], the Carbapenem-Inactivation Method[9], detection

[1]Institute of Medical Microbiology and Virology, Carl von Ossietzky University Oldenburg, Oldenburg, Germany. [2]Department of Ecology, School of Biology/Chemistry, University of Osnabrück, Osnabrück, Germany. [3]Department of Medical Microbiology and Infection Prevention, University Medical Center Groningen, University of Groningen, Groningen, The Netherlands. [4]Institute for Medical Microbiology, Immunology and Hygiene, University Hospital Cologne and Faculty of Medicine, University of Cologne, Cologne, Germany. [5]Department of Machine Learning and Systems Biology, Max Planck Institute of Biochemistry, Martinsried, Germany. [6]Institute of Medical Microbiology, University of Zurich, Zurich, Switzerland. [7]Team Resist UMR1184 Immunology of Viral, Auto-Immune, Hematological and Bacterial diseases (IMVA-HB), INSERM, Faculty of Medicine, Université Paris-Saclay, CEA, LabEx LERMIT, Le Kremlin-Bicêtre, France. [8]Associated French National Reference Center for Antibiotic Resistance: Carbapenemase-Producing Enterobacteriaceae, Le Kremlin-Bicêtre, France. [9]German Centre for Infection Research, Partner Site Bonn-Cologne, Cologne, Germany. ✉e-mail: axel.hamprecht@uni-oldenburg.de

of carbapenem hydrolysis with MALDI-TOF mass spectrometry[10,11] or lateral flow assays[12]. Since all these confirmatory tests require additional time, material and workforce, an efficient preselection of isolates for further testing is crucial.

Screening for CPEs as proposed by EUCAST using meropenem and/or ertapenem can be challenging, as there are certain carbapenemase types and species that are harder to detect than others. Especially OXA-48-like carbapenemase producers are often susceptible to meropenem and may thus be missed via a simple meropenem screening[13]. Carbapenemase-producing *Proteus mirabilis* are harder to detect due to increased carbapenem susceptibility[14] and unique carbapenemase variants that are less frequent in other *Enterobacterales*, such as $bla_{OXA-23}$. While species-specific algorithms have been established for *P. mirabilis*[14], a universal algorithm would be more effective for use in the routine laboratories.

The Antibiogram Committee of the French Society of Microbiology (CA-SFM) proposed a CPE screening algorithm using three antibiotic disks (ceftazidime-avibactam, temocillin, and a carbapenem—either meropenem, imipenem, or ertapenem)[15]. This algorithm was tested by Duque et al. in 2024 on a collection of 518 isolates, yielding 97.8% sensitivity and 45.5% specificity[1,16]. For diagnostic algorithms, there is an inevitable tradeoff between sensitivity and specificity, and most current algorithms favor high sensitivity over specificity, increasing the amounts of unnecessarily tested false positive isolates in the laboratory.

In recent years, artificial Intelligence (AI) and especially machine learning has gained traction in aiding scientists and clinicians to interpret data and resolve patterns that may not be recognizable at first sight. There have been several attempts to use AI to improve carbapenemase detection, e.g., by using ChatGPT to analyse inhibition zones[17], or by using it to identify peaks in MALDI-TOF MS spectra indicating the presence of a carbapenemase or a specific resistance[18,19]. However, detection of resistance using MALDI-TOF MS is usually an indirect method and requires large training datasets to reach acceptable sensitivities. Susceptibility testing using disk diffusion is an established, widely available method, which is performed in many clinical microbiology laboratories worldwide, and the data is thus readily available.

In this study, we therefore aim to develop an optimized screening approach by analyzing inhibition zone diameters of eight antibiotics on a collection of 385 clinical *Enterobacterales* isolates using a machine learning based model.

## Results
### Performance of existing screening algorithms on our dataset
We applied both the CA-SFM algorithm and the EUCAST screening cut-off to the 385 isolates in our dataset. The CA-SFM yielded a sensitivity of 95.0% (CI: 91.4–97.4%) and a specificity of 40.1% (CI: 32.2–48.5%), leading to a Youden index of 0.351 (Table 1). The number of negative isolates that would be sent for confirmatory testing after the screening was 88 out of 147 negative isolates, meaning that 59.9% of negative isolates would be further tested. This is 28.0% of all samples sent for confirmatory testing.

Application of EUCAST screening resulted in a slightly higher sensitivity of 97.9% (CI: 95.2–99.3%), but a lower specificity of 8.2% (CI: 4.3–13.8%). This would result in 135 negative isolates being unnecessarily tested with a confirmatory test, which is 91.8% of all negative isolates, and 36.7% of all samples sent for confirmatory testing.

### Development of an algorithm
Based on the inhibition zone diameters of the eight antibiotics tested, we first built a simple decision tree based on only these diameters and the species. Using nested cross-validation, this yielded a model with 89.5% sensitivity (CI: 84.9–93.1%) and 86.4% specificity (CI: 79.8–91.5%) (Table 2).

To allow for a more accurate prediction, we trained a random forest model on the internal dataset. This model showed an increased sensitivity of 92.9% (CI: 88.8–95.8) and a specificity of 86.4% (CI: 79.8–91.5%). In addition, this model type can also calculate the probability of each classification, allowing users to adjust the cut-off and adjust the model towards a higher sensitivity.

To allow for a more robust model, the inhibition zone diameters were used to create new variables, i.e., the difference between each pair of inhibition zone diameters. The random forest model including the additional engineered features (random forest expanded) yielded 95.4% sensitivity (CI: 91.9–97.7%) and 87.8% specificity (CI: 81.3–92.6%). The additional modification of the custom threshold of 0.5 to 0.6 alters the translation of probability predictions into classification, adjusting the resulting model towards a higher sensitivity while decreasing specificity. The resulting model (random forest expanded sensitive = CarbaDetector) predicts carbapenemase production with a sensitivity of 96.6% (CI: 93.5–98.5%) and a specificity of 85.0% (CI: 78.2–90.4%), resulting in only 8.7% of isolates that need to be further tested being false positives, which is a significant decrease when compared to the EUCAST and CA-SFM algorithm (Supplementary Data 1).

For the CarbaDetector model that was trained on the whole internal dataset for use on external datasets, features with the highest importance as determined by mean decreased accuracy are imipenem-relebactam, imipenem, temocillin, the difference between temocillin and ceftazidime-avibactam, as well as the difference between ertapenem and imipenem-relebactam (see Fig. 1). While the species of an isolate has an impact on the outcome of the prediction, the difference between species does not warrant the construction of species-specific models (see Supplementary Information) (Fig. 2).

The nested cross-validation estimate on the internal dataset resulted in eight false negative isolates. These included isolates producing VIM-1 ($n = 3$), OXA-244 ($n = 2$), IMP-13, OXA-181, and KPC-3 ($n = 1$ each), with 6/8 isolates being susceptible to at least one carbapenem by EUCAST breakpoints, Table S2. For imipenem-relebactam all isolates' inhibition zones were larger than 22 mm (corresponding to susceptibility).

### Validation of CarbaDetector using external datasets
The performance of CarbaDetector was assessed using two external datasets: (i) a dataset tested with all eight antibiotics (external dataset A), and (ii) a dataset provided by Duque et al. with only five of the eight measurements (external dataset B). To predict the presence of carbapenemases for external dataset B, the available five inhibition zone measurements were used to impute the missing three values (based on the internal training dataset). Finally, the predictions were performed using the available and the imputed values. CarbaDetector achieved a sensitivity of 96.3% (CI: 89.4–99.2%) and a specificity of 86.1% (CI: 80.6–90.6%) on dataset A, which was higher than the values obtained by the CA-SFM algorithm and the EUCAST screening cut-off on this dataset (Table 3).

When assessing the performance of CarbaDetector on the external dataset B, the CarbaDetector prediction algorithm yielded a sensitivity of 91.2% (CI: 87.8–93.9%) and a specificity of 87.0% (CI: 80.7–91.9%). CarbaDetector showed increased specificity when compared to the CA-SFM algorithm and EUCAST screening, with a better sensitivity than the EUCAST algorithm, but decreased sensitivity compared to the CA-SFM algorithm when imputing three out of eight measures (Table 3). The highest Youden index was achieved by CarbaDetector, both with eight and five inhibition zone diameters.

### The CarbaDetector web-app
Using the validated CarbaDetector model, we created the web-app CarbaDetector, which can be found at https://uol.de/carba-detector. Here, the user can enter the inhibition zone diameters measured for their isolate (Fig. 2). Based on these measurements, the web-app predicts the probability that an isolate is a carbapenemase producer and informs the user in real-time.

## Discussion

We investigated and tested current screening methods for CPE prediction and aimed to develop a machine learning tool to accurately predict the presence of carbapenemases in *Enterobacterales* using simple inhibition zone diameters.

EUCAST has established cut-offs that trigger further carbapenemase testing, based on meropenem and ertapenem inhibition zones or minimal inhibitory concentrations[4]. When applying these cut-offs to our dataset, a high sensitivity of 97.9% (CI: 95.2–99.3%) is achieved, but with a very low specificity of 8.2% (CI: 4.3–13.8%). CarbaDetector performs with a similar sensitivity of 96.6% (CI: 93.5–98.5%), but a significantly higher specificity of 85.0% (CI: 78.2–90.4%), resulting in an ~10-fold increase in specificity, leading to a 6-fold decrease of negative isolates that require confirmatory tests. This decrease of resources might be an important consideration especially in resource-limited settings. Moreover, even on an external dataset with missing values, CarbaDetector performed with 91.2% sensitivity and 87.0% specificity, showing its applicability even to incomplete data sets that miss some of the essential antibiotic test results. Even with missing values, the sensitivity was only slightly lower than that of the CA-SFM algorithm, but the specificity and the Youden index higher.

For dataset A, CarbaDetector predicted 28 isolates to be negative that both the EUCAST screening algorithm and the CA-SFM algorithm wrongfully marked as positive. These 28 isolates were mostly *K. pneumoniae* isolates (n = 15), followed by *E. coli* (n = 6), *Enterobacter cloacae* complex (n = 5), *C. freundii* (n = 1) and *K. aerogenes* (n = 1). All of these isolates showed an inhibition zone diameter smaller than 25 mm for ertapenem which leads to them being flagged by the EUCAST screening algorithm. There were two carbapenemase-producing isolates that were missed by all three screening algorithms. These were a *K. pneumoniae* and a *P. mirabilis* isolate, both harboring $bla_{NDM-1}$. Since CarbaDetector presents a probability score with each prediction, the user can however, increase sensitivity by defining their own probability cutoff. Despite its promises, AI has not been extensively used for carbapenemase detection. Recently, a GPT agent has demonstrated potential for the prediction of ESBL, AmpC and carbapenemases based on inhibition zone measurements, adhering to the expert rules given by EUCAST. However, the GPT agent showed lower specificity in some resistance mechanisms compared to clinical microbiologists, leading to more additional tests[17].

There are some limitations to our study. Even though the algorithm was tested on datasets from three different countries, the performance of CarbaDetector should be assessed using more and diverse isolates of different geographical origins and resistance mechanisms. Additionally, so far it includes inhibition zones of some antibiotics that are likely not included in all test panels (e.g., temocillin, imipenem-relebactam). Nevertheless, the model can be applied even if not all eight antibiotics have been tested, since missing diameters are imputed based on the underlying dataset. Importantly, the CarbaDetector is not IVDR (In Vitro Diagnostic Regulation) conform, and is for research use only. However, since we did not only use one dataset, but two completely independent external datasets for validation, we expect CarbaDetector to perform well when put to practice. It has to be considered that the control group was composed to include a very high proportion of challenging isolates, with increased carbapenem MICs but without carbapenemases. In the routine setting, all algorithms will likely achieve a higher performance than in this demanding strain set.

We chose disk diffusion as the susceptibility testing method for this pilot study, since it is commonly used worldwide and recommended by both EUCAST and CLSI. Additionally, it has the advantage of providing a wide range of quantitative inhibition zone data, which could be used for developing the model. Semi-automated susceptibility testing systems are also commonly used nowadays, but often have a limited calling range of minimal inhibitory concentrations. Nevertheless, further development of CarbaDetector will include MIC data from other susceptibility testing systems. One challenge when working with disk diffusion is that specific combinations of species and antibiotics can lead to difficult to interpret results, e.g., due to swarming or the formation of microcolonies in the inhibition zones. For this study, we adhered to EUCAST guidelines for disk diffusion methodology[20]. For optimal results, it is recommended to follow these standards when using CarbaDetector.

For training the algorithm, we used an isolate collection that consists of isolates from German centers and including isolates with high, medium and low carbapenem MICs. The species and carbapenemases are therefore not evenly distributed. It is important to note that this can affect the model outcome and bias the algorithm. The collection we chose is, however, representative of what would be

**Table 1 | Evaluation of the CA-SFM algorithm and the EUCAST screening cut-off using our dataset (n = 385 isolates)**

|  | CA-SFM algorithm | EUCAST screening |
|---|---|---|
| Sensitivity | 95.0% (91.4–97.4%) | 97.9% (95.2–99.3%) |
| Specificity | 40.1% (32.2–48.5%) | 8.2% (4.3–13.8%) |
| Youden index | 0.351 | 0.061 |

**Table 2 | Evaluation of the different trained models using the internal dataset (n = 385) for validation using nested cross-validation**

|  | Decision tree | Random forest basic[a] | Random forest expanded[b] | Random forest expanded sensitive (CarbaDetector) |
|---|---|---|---|---|
| Sensitivity | 89.5% (84.9–93.1%) | 92.9% (88.8–95.8%) | 95.4% (91.9–97.7%) | 96.6% (93.5–98.5%) |
| Specificity | 86.4% (79.8–91.5%) | 86.4% (79.8–91.5%) | 87.8% (81.3–92.6%) | 85.0% (78.2–90.4%) |
| Youden index | 0.759 | 0.793 | 0.832 | 0.816 |
| Isolates requiring complementary tests | 233 (60.5%) | 241 (62.6%) | 245 (63.6%) | 252 (65.5%) |
| Negative isolates requiring complementary tests | 20 (13.6%) | 20 (13.6%) | 18 (12.2%) | 22 (15.0%) |
| True positive | 213 | 221 | 227 | 230 |
| True negative | 127 | 127 | 129 | 125 |
| False positive | 20 | 20 | 18 | 22 |
| False negative | 25 | 17 | 11 | 8 |

[a]*Basic* refers to the models based on only the inhibition zone measurements and the species.
[b]*Expanded* refers to the datasets including the differences in inhibition zones of the eight antibiotics.

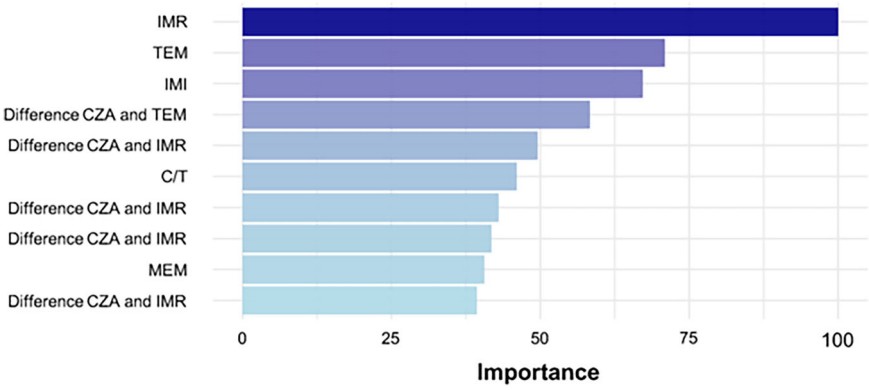

**Fig. 1 | Variable importance plot of the ten most important variables in the CarbaDetector model.** IMR imipenem-relebactam, TEM temocillin, IMI imipenem, CZA ceftazidime-avibactam, C/T Ceftolozan/Tazobactam, ETP ertapenem, MEM meropenem. "Difference" describes the difference of inhibition zone diameters for two antibiotic disks for the same isolate. See Table S1 for all variable importance parameters.

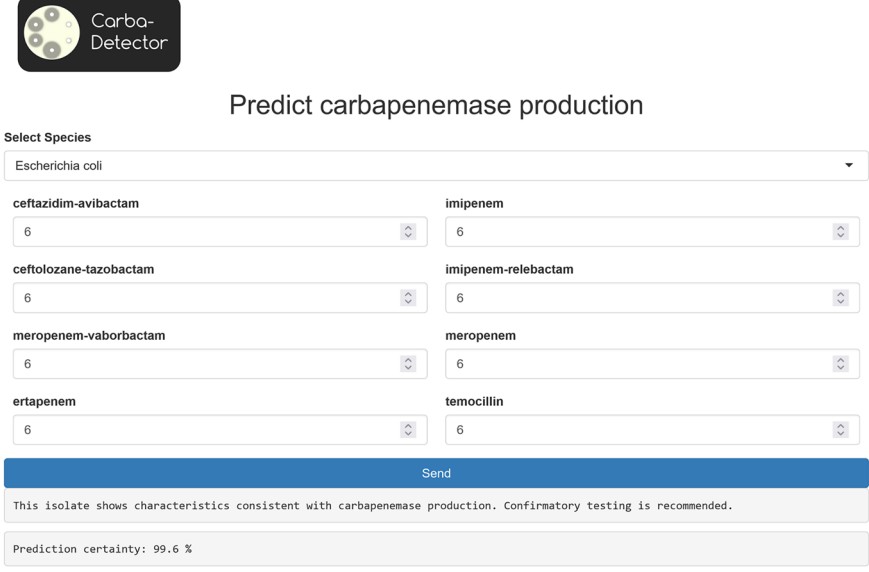

**Fig. 2 | Interface of the CarbaDetector web-app.**

**Table 3 | Performance of the CarbaDetector on two external datasets, with dataset A containing inhibition zones for all eight antibiotics, dataset B for only five (missing values were imputed by CarbaDetector)**

| | dataset A | | | dataset B | | |
|---|---|---|---|---|---|---|
| | **Carba-Detector** | **CA-SFM algorithm** | **EUCAST** | **Carba-Detector** | **CA-SFM algorithm** | **EUCAST** |
| Sensitivity | 96.3% (89.4–99.2%) | 95.0% (87.7–98.6%) | 95.0% (87.7–98.6%) | 91.2% (87.8–93.9%) | 97.8% (95.7–99.0%) | 87.1% (83.2–90.4%) |
| Specificity | 86.1% (80.6–90.6%) | 73.3% (66.6–79.2%) | 37.6% (30.9–44.7%) | 87.0% (80.7–91.9%) | 45.5% (37.4–53.7%) | 39.0% (31.2–47.1%) |
| Youden index | 0.824 | 0.683 | 0.326 | 0.782 | 0.433 | 0.261 |
| Isolates requiring complementary tests | 105 (37.2%) | 130 (46.1%) | 202 (71.6%) | 352 (68.0%) | 440 (84.9%) | 408 (78.7%) |
| Negative isolates requiring complementary tests | 28 (13.9%) | 54 (27.8%) | 126 (62.4%) | 20 (13.0%) | 84 (54.5%) | 94 (61.0%) |
| True positive | 77 | 76 | 76 | 332 | 356 | 317 |
| True negative | 174 | 148 | 76 | 134 | 70 | 60 |
| False positive | 28 | 54 | 126 | 20 | 84 | 94 |
| False negative | 3 | 4 | 4 | 32 | 8 | 47 |

encountered in a routine clinical setting in Germany, with predominance of OXA-48-like carbapenemases. Since the model was additionally evaluated with two external datasets, we are optimistic that this approach yielded the best balance between keeping as many isolates as possible in the training dataset, leading to more accuracy, and striving for little bias.

In conclusion, CarbaDetector is, to the best of our knowledge, the first AI-based open-access web-app to predict the production of carbapenemases in *Enterobacterales* isolates based on inhibition zones. The model combines the accuracy of sophisticated data analysis with the applicability of a simple algorithm and is thus a powerful tool in *Enterobacterales* analysis. It is easy-to-use and its application in the routine laboratory could help make informed decisions on whether or not isolates need to be further tested for carbapenemases.

For the further development, we will include more isolates with different species and resistance mechanisms, making the prediction even more accurate. With a larger collection including a greater variety of species and carbapenemase types including isolates from different locations globally, we are confident that future versions of CarbaDetector can also be applied to predict which category carbapenemase is present in an isolate, potentially further decreasing the need for confirmatory tests. Moreover, we are planning to include more antibiotics in future testing of the isolates, to further improve usability of the model.

## Methods
### Strain collection
This study comprised 385 non-duplicate clinical *Enterobacterales* isolates, collected from 2012 to 2021 at the University Hospital Cologne and Klinikum Oldenburg in routine diagnostics. Species identity was determined using MALDI-TOF mass spectrometry and confirmed by whole genome sequencing (WGS). Of all isolates, 238 (61.8%) were carbapenemase producers, 147 (38.2%) were carbapenemase-negative. Molecular characterization of all isolates was performed by WGS on the Illumina platform, as previously described[21]. Briefly, DNA was extracted from pure bacterial cultures using the DNeasy UltraClean Microbial Kit (Qiagen, Hilden, Germany). Whole genome sequencing was performed by Novogene (Beijing, China). Genomic DNA libraries were prepared with the Novogene NGS DNA Library Prep Set with an average insert size of 350 bp, followed by paired-end 150 bp sequencing on an Illumina NovaSeq platform (Illumina, San Diego, CA, USA). Presence or absence of carbapenemase genes was confirmed using ResFinder v4.7.2[22,23]. The results of molecular characterization were used as reference standard to evaluate the algorithm performance. Six species constituted 88.8% of the isolates, namely *K. pneumoniae*, *E. coli*, *C. freundii*, *E. cloacae*, *P. mirabilis* and *S. marcescens*. The most frequent carbapenemase group present was $bla_{OXA-48-like}$ (46.6%). Detailed characteristics of isolates and datasets is provided in the Supplementary Information.

### Susceptibility testing
Susceptibility testing was performed at the Institute of Medical Microbiology and Virology, University Oldenburg according to EUCAST standards[20], employing disks containing meropenem, ertapenem, imipenem, meropenem-vaborbactam, ceftazidime-avibactam, ceftolozane-tazobactam, temocillin (Oxoid, Basingstoke, UK), and imipenem-relebactam (Mast Group, Merseyside, UK) on Mueller-Hinton agar (Oxoid, Basingstoke, UK). Inhibition zones were measured manually.

### Assessing the performance of the novel CA-SFM algorithm and the EUCAST screening process
To set the baseline for our model, we assessed the CA-SFM algorithm and the EUCAST screening algorithm for carbapenemase detection by applying it to all three datasets, using WGS results as ground truth. To develop a universal algorithm using R (*rpart* (4.1.24) and *RandomForest* (4.7.1.2) packages[24,25]), we built a

decision tree and a random forest model using (i) species and the standard-scaled inhibition zone diameters and (ii) additionally a random forest model using the scaled differences in inhibition zone diameters. The differences in inhibition zone diameters (instead of only the raw diameters) were included once per antibiotics pair in order to compensate for laboratory-specific differences between measurements. To increase sensitivity, several cutoffs (0.5, 0.6, 0.7, 0.75) in the random forest model classification were assessed, with the final cutoff being 0.6, meaning that samples were predicted as "negative", if a probability of more than 60% was determined, as opposed to the default 50%.

To estimate model performance, we employed nested cross-validation with 10 outer and 10 inner folds using the *nestedcv* R package[26]. Where possible, sampling was stratified for species and presence of carbapenemase genes. Class weights were applied to address the imbalanced distribution between carbapenemase negative and positive samples.

After estimating the performance on our own dataset (Supplementary Data 1), the final model was trained on the whole dataset with hyperparameter tuning via 10-fold cross-validation and applied to the external datasets for additional validation.

### Validation of our algorithm using external datasets
To further validate the trained model and its correct prediction of CPE, the resulting model (CarbaDetector) has been used firstly to predict carbapenemase production on a set of 282 *Enterobacterales* isolates from Switzerland (University of Zurich) with and without carbapenemase production (external dataset A, included in Supplementary Data 2). For this dataset, inhibition zone diameters for all eight antibiotics used in the algorithm were determined.

Secondly, prediction of carbapenemase production on incomplete datasets (where not all eight recommended antibiotic disks were used) was tested on a different, previously published dataset containing the disk diffusion diameters of 518 *Enterobacterales* isolates submitted for carbapenemase testing to the French reference laboratory for multidrug-resistant Gram-negatives (external dataset B, included in Supplementary Data 3, originally used for the assessment of the CA-SFM algorithm[16]). Here, the diameters were measured using SIRscan and verified manually. Using the inhibition zone diameters for ertapenem, meropenem, imipenem, temocillin, and ceftazidim-avibactam, we imputed the missing values for imipenem-relebactam, meropenem-vaborbactam, and ceftolozane-tazobactam based on our dataset applying the *missRanger* R package[27]. Then, using the built model, the presence or absence of carbapenemase production was predicted. Information on statistical analyses and the development of the app can be found in the Supplementary Information.

### Ethics approval
The bacterial strains were isolated during routine diagnostics and anonymized. As no patient data were analyzed, ethical approval was not required for this type of study according to §15 of the professional code for physicians.

### Reporting summary
Further information on research design is available in the Nature Portfolio Reporting Summary linked to this article.

## Data availability
The inhibition zone measurement data as well as application of EUCAST, CA-SFM and CarbaDetector are available as source datasets and included in the Supplementary Information.

## Code availability
Model code is available in the associated CodeOcean capsule: https://codeocean.com/capsule/8305077/tree.

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

## Acknowledgements

We thank Dr. Chantal Quiblier, Natalia Kolesnik-Goldmann, Yukino Gütlin and Natalia Kolesnik-Goldmann from the Institute of Medical Microbiology at the University of Zurich for extraction of data.

## Author contributions

Conceptualization: L.K.M., A.H. Methodology: L.K.M., J.S. Software: L.K.M. Formal Analysis: L.K.M. Investigation: C.C., L.T. Resources: A.H., O.N., L.D., R.A.B., A.E. Writing—Original Draft: L.K.M., C.C., A.H. Writing—Review and Editing: L.K.M., C.C., J.S., O.N., L.D., R.A.B., A.E., A.H. Visualization: L.K.M. Supervision: A.H. Funding Acquisition: A.H.

## Funding

## Competing interests

The authors declare no competing interests.
