## [Peer Review File · Nature Communications]

CarbaDetector: A Machine Learning Model for Detecting Carbapenemase-Producing Enterobacterales from Disk Diffusion Tests

Corresponding Author: Professor Axel Hamprecht

Version 0:

Reviewer comments:

Reviewer #1

(Remarks to the Author)

The manuscript titled “CarbaDetector: A Machine Learning Model for Detecting Carbapenemase-Producing Enterobacterales from Disk Diffusion Tests” presents a machine learning (ML) model based on random-forest algorithms for predicting carbapenemase production in Enterobacterales, using inhibition zone diameters from eight antibiotics.

The topic is highly relevant, and the study design is clearly and rigorously described. The datasets used to train and validate the model appear robust and provide a reliable benchmark. Notably, the model demonstrates remarkably high specificity, as reflected by a Youden index greater than 0.750, which is impressive given the generally low specificity of currently used algorithms, resulting in increased workload due to false positives.

The manuscript is well written and methodologically sound. I do not have major criticisms. However, I offer the following comments for consideration in the final revision of the manuscript:

1. Several elements currently presented in the Supplementary Materials are crucial for understanding the study design and should be integrated into the main text. For instance, lines 104–112 would benefit from further elaboration by incorporating relevant details currently provided only in the supplementary section.
2. Figure S2, which illustrates the importance of different variables in the model’s predictions, is highly informative and should be discussed in the main manuscript. This would enhance the reader’s understanding of how each antibiotic contributes to the decision process of the ML model.
3. It is well known that the expression of resistance genes can vary significantly across bacterial species. For example, a KPC carbapenemase shows a different level of resistance if present in *Klebsiella pneumoniae* (usually high level) or *Escherichia coli* (usually, low level). I am curious whether the model accounts for this variability. I noticed that the publicly available (<https://uol.de/carba-detector>) requires the user to input the bacterial species: does this input influence the algorithm’s decision-making process?
4. The six isolates reported as false negatives are indeed challenging cases. However, when their values are entered into the web-based application, not all of them are classified as carbapenemase-negative (e.g., see *E. coli* B2-3 in the reviewer attachments). Could the authors clarify this discrepancy between the tool’s output and the reported results?
5. Among the limitations, it should be noted that the interpretation of inhibition zones can be difficult in certain species-antibiotic combinations, for instance, due to the presence of microcolonies. In order to allow the best performances of the system, the authors should consider make aware the readers of the need for correct interpreting ambiguous zones.

(Remarks on code availability)

Reviewer #2

(Remarks to the Author)

In the introduction, the authors should include in their rationale a brief justification for why a Machine Learning (ML) model based on disk diffusion test data is preferred or advantageous over other high-resolution approaches such as whole-genome sequencing (WGS)

Methods

The authors should clearly state whether ethical approval and clearance were obtained for this study, especially since it involves the use of data derived from clinical specimens. Even when anonymized or retrospective, the use of patient-derived microbiological data typically requires approval from an institutional review board or ethics committee to ensure compliance with ethical standards and data protection regulations. Please include the name of the approving body and the reference number of the ethical clearance, if applicable.

Line 104: The manuscript states that 385 isolates were used for training the model, but there is no breakdown of the species composition within this dataset. This information is critical, as the performance of carbapenemase-producing Enterobacterales (CPE) prediction models can vary significantly across different species due to intrinsic differences in resistance mechanisms and phenotypic profiles. Please provide a detailed breakdown of the species included in the training set and discuss how species distribution may have influenced model performance and generalizability.

Line 108: Molecular characterization of all isolates was performed by WGS on the Illumina platform, as previously described. It is important to briefly include details of the platform and the sequencing protocol here

Line 110: ResFinder, please include the version

Line 113: Please include the name of the testing laboratory for AST and Whole-genome sequencing

Line 122: The authors mention that they "built a decision tree and a random forest model" for their analysis. Could the authors clarify why only these two models were chosen? It would strengthen the study to explain whether other machine learning algorithms (e.g., support vector machines, gradient boosting, logistic regression, or neural networks) were considered, and if not, why. Ideally, model selection should involve comparing multiple algorithms to identify the best-performing approach based on relevant performance metrics. Please elaborate on the rationale for limiting the analysis to these two models.

Line 130: nestedcv R package. Please include the reference

Line 86: With a bigger collection, we are confident that future versions of CarbaDetector can also be applied to predict which category carbapenemase is present in an isolate, potentially further decreasing the need for confirmatory tests. Could the authors clarify what is meant by a "bigger collection"? Does this refer to a larger number of isolates overall, increased species diversity, more comprehensive phenotypic or genotypic data, or inclusion of isolates with confirmed carbapenemase types? Providing more detail would help readers understand what additional data is needed to enhance the model's predictive capabilities.

(Remarks on code availability)

Version 1:

Reviewer comments:

Reviewer #1

(Remarks to the Author)

The manuscript titled "CarbaDetector: A Machine Learning Model for Detecting Carbapenemase-Producing Enterobacterales from Disk Diffusion Tests" presents a machine learning (ML) model based on random-forest algorithms for predicting carbapenemase production in Enterobacterales, using inhibition zone diameters from eight antibiotics.

The manuscript is well written and methodologically sound. In this revised version of the paper the authors complied with the suggestions of the reviewers and now the manuscript appears further improved.

As a only remark, line 164: "according to §15 of the professional". There is probably a typo, please correct it.

(Remarks on code availability)

Reviewer #2

(Remarks to the Author)

Thank you for the thorough and thoughtful revision. You have satisfactorily addressed all of my previous comments.

(Remarks on code availability)

Point-by-point response Nature Communications manuscript

We thank the the reviewers for their helpful comments, which have further improved the manuscript. We addressed all reviewers' comment and further refined the model.

REVIEWER COMMENTS

Reviewer #1 (Remarks to the Author):

The manuscript titled "CarbaDetector: A Machine Learning Model for Detecting Carbapenemase-Producing Enterobacterales from Disk Diffusion Tests" presents a machine learning (ML) model based on random-forest algorithms for predicting carbapenemase production in Enterobacterales, using inhibition zone diameters from eight antibiotics.

The topic is highly relevant, and the study design is clearly and rigorously described. The datasets used to train and validate the model appear robust and provide a reliable benchmark. Notably, the model demonstrates remarkably high specificity, as reflected by a Youden index greater than 0.750, which is impressive given the generally low specificity of currently used algorithms, resulting in increased workload due to false positives.

The manuscript is well written and methodologically sound. I do not have major criticisms. However, I offer the following comments for consideration in the final revision of the manuscript:

1. Several elements currently presented in the Supplementary Materials are crucial for understanding the study design and should be integrated into the main text. For instance, lines 104–112 would benefit from further elaboration by incorporating relevant details currently provided only in the supplementary section.

We now added information on species distribution and carbapenemase types (Lines 115-117). Due to word count constraints, it is unfortunately not possible to include all information in the main manuscript.

2. Figure S2, which illustrates the importance of different variables in the model's predictions, is highly informative and should be discussed in the main manuscript. This would enhance the reader's understanding of how each antibiotic contributes to the decision process of the ML model.

We moved this figure (now Figure 1 in the main text) to the main part and included an exhaustive list of all importance parameters in the supplementary material.

3. It is well known that the expression of resistance genes can vary significantly

across bacterial species. For example, a KPC carbapenemase shows a different level of resistance if present in *Klebsiella pneumoniae* (usually high level) or *Escherichia coli* (usually, low level). I am curious whether the model accounts for this variability. I noticed that the publicly available (<https://uol.de/carba-detector>) requires the user to input the bacterial species: does this input influence the algorithm's decision-making process?

Yes, the species is taken into account in our model. However, in the current model version, the species as a predictor is of very little importance. This is partly due to the uneven distribution of carbapenemase types between species. Since this difference is well known and we are planning to expand the database and thus update the model, we included "species" as a predictor already in this model.

4. The six isolates reported as false negatives are indeed challenging cases. However, when their values are entered into the web-based application, not all of them are classified as carbapenemase-negative (e.g., see *E. coli* B2-3 in the reviewer attachments). Could the authors clarify this discrepancy between the tool's output and the reported results?

This is indeed the case. In our manuscript, we report two different processes: First, we develop a suitable model to predict the presence of carbapenemases based on our internal dataset. For this, we perform nested cross-validation, meaning that there are several rounds of partitioning the dataset into training and testing datasets. The final model in this process is thus a model that is built on a fraction of the internal dataset and validated also using a fraction of the internal dataset.

In the second process, using the hyperparameters obtained in the first process, we use the entire internal dataset to build the CarbaDetector model and validate it using the external datasets. Here, we use the entire internal dataset, instead of a fraction. Therefore, the resulting model is slightly different (and the previously reported challenging isolates are part of the training dataset), so that a repeated testing of these isolates can lead to different results.

5. Among the limitations, it should be noted that the interpretation of inhibition zones can be difficult in certain species-antibiotic combinations, for instance, due to the presence of microcolonies. In order to allow the best performances of the system, the authors should consider make aware the readers of the need for correct interpreting ambiguous zones.

Thank you for this comment. Indeed, there are specific combinations of pathogens and antibiotics where it is crucial to follow established guidelines. We added a sentence explaining the guidelines we adhered to and emphasized that following these guidelines is crucial to obtain meaningful predictions (lines 286-290).

Reviewer #2 (Remarks to the Author):

In the introduction, the authors should include in their rationale a brief justification for why a Machine Learning (ML) model based on disk diffusion test data is preferred or

advantageous over other high-resolution approaches such as whole-genome sequencing (WGS)

Thank you, we included that often disk diffusion susceptibility testing is routinely performed, so that the data is readily available (Lines 97-99). Moreover, as stated in the discussion, the app is supposed to improve the selection of isolates for confirmatory testing (e.g. by PCR or WGS), not replace it.

Methods

The authors should clearly state whether ethical approval and clearance were obtained for this study, especially since it involves the use of data derived from clinical specimens. Even when anonymized or retrospective, the use of patient-derived microbiological data typically requires approval from an institutional review board or ethics committee to ensure compliance with ethical standards and data protection regulations. Please include the name of the approving body and the reference number of the ethical clearance, if applicable.

The bacterial strains were isolated during routine diagnostics and anonymized. As no patient data were analyzed, ethical approval was not required for this type of study according to §15 of the professional code for physicians. We have included this information in the revised manuscript (Lines 161-164).

Line 104: The manuscript states that 385 isolates were used for training the model, but there is no breakdown of the species composition within this dataset. This information is critical, as the performance of carbapenemase-producing Enterobacterales (CPE) prediction models can vary significantly across different species due to intrinsic differences in resistance mechanisms and phenotypic profiles. Please provide a detailed breakdown of the species included in the training set and discuss how species distribution may have influenced model performance and generalizability.

An exact breakdown of the included species can be found in the supplementary materials. We have added a paragraph discussing the impact that species and an imbalance in species in the training dataset can have on the final model performance (Lines 291-298).

Line 108: Molecular characterization of all isolates was performed by WGS on the Illumina platform, as previously described. It is important to briefly include details of the platform and the sequencing protocol here

The following details were added to the methods (Lines 109-113):

“DNA was extracted from pure bacterial cultures using the DNeasy UltraClean Microbial Kit (Qiagen, Hilden, Germany). Whole genome sequencing was performed by Novogene (Hongkong, China). Genomic DNA libraries were prepared with the Novogene NGS DNA Library Prep Set with an average insert size of 350 bp, followed

by paired-end 150 bp sequencing on an Illumina NovaSeq platform (Illumina, San Diego, CA, USA).”

Line 110: ResFinder, please include the version

We now included the version we used for analysis.

Line 113: Please include the name of the testing laboratory for AST and Whole-genome sequencing

AST was performed at the Institute of Medical Microbiology and Virology, University of Oldenburg. WGS was done by Illumina sequencing at Novogene; Beijing, China. This information was added in line 111 and 120.

Line 122: The authors mention that they "built a decision tree and a random forest model" for their analysis. Could the authors clarify why only these two models were chosen? It would strengthen the study to explain whether other machine learning algorithms (e.g., support vector machines, gradient boosting, logistic regression, or neural networks) were considered, and if not, why. Ideally, model selection should involve comparing multiple algorithms to identify the best-performing approach based on relevant performance metrics. Please elaborate on the rationale for limiting the analysis to these two models.

We selected decision tree and random forest algorithms based on the critical requirement for interpretability in the setting of clinical screenings. Unlike black-box models such as neural networks or svm, or hard-to-interpret sequential tree algorithms like gradient boosting, random forests provide comparatively transparent decision pathways. While more complex models could yield marginally better results in terms of model performance, we think it is essential that the prediction and the rules the algorithm follows can be understood and traced by clinicians and researchers. While this may come at the expense of better results, it was a conscious choice in order to both garner trust of potential users.

Line 130: nestedcv R package. Please include the reference

We included the reference [REF 26].

Line 86: With a bigger collection, we are confident that future versions of CarbaDetector can also be applied to predict which category carbapenemase is present in an isolate, potentially further decreasing the need for confirmatory tests. Could the authors clarify what is meant by a "bigger collection"? Does this refer to a larger number of isolates overall, increased species diversity, more comprehensive phenotypic or genotypic data, or inclusion of isolates with confirmed carbapenemase

types? Providing more detail would help readers understand what additional data is needed to enhance the model's predictive capabilities.

Thank you very much for this insightful comment. A combination of the factors you mentioned would be perfect, but we are of course limited to the isolates that are available and can be made available in the future. There are a number of factors that we would want to improve on the model in the future. In the immediate future, we are planning to expand the collection with more isolates from different locations, more diverse species and we aim to stratify the uneven distribution of carbapenemase classes, which is caused by inclusion of isolates from Europe only. We added a paragraph in the discussion to explain this aspect (Lines 305-311).

Point-by-point response Nature Communications manuscript

REVIEWER COMMENTS

Reviewer #1 (Remarks to the Author):

The manuscript titled “CarbaDetector: A Machine Learning Model for Detecting Carbapenemase-Producing Enterobacterales from Disk Diffusion Tests” presents a machine learning (ML) model based on random-forest algorithms for predicting carbapenemase production in Enterobacterales, using inhibition zone diameters from eight antibiotics.

The manuscript is well written and methodologically sound. In this revised version of the paper the authors complied with the suggestions of the reviewers and now the manuscript appears further improved.

As a only remark, line 164: “according to §15 of the professional”. There is probably a typo, please correct it.

checked

Reviewer #2 (Remarks to the Author):

Thank you for the thorough and thoughtful revision. You have satisfactorily addressed all of my previous comments.

We thank the reviewer for this positive comment.